Amended: Addendum

# Observation of an anti-PT-symmetric exceptional point and energy-difference conserving dynamics in electrical circuit resonators

Youngsun Choi[1], Choloong Hahn[2], Jae Woong Yoon[1,3] & Seok Ho Song[1]

Parity-time (PT) symmetry and associated non-Hermitian properties in open physical systems have been intensively studied in search of new interaction schemes and their applications. Here, we experimentally demonstrate an electrical circuit producing key non-Hermitian properties and unusual wave dynamics grounded on anti-PT (APT) symmetry. Using a resistively coupled amplifying-*LRC*-resonator circuit, we realize a generic APT-symmetric system that enables comprehensive spectral and time-domain analyses on essential consequences of the APT symmetry. We observe an APT-symmetric exceptional point (EP), inverse PT-symmetry breaking transition, and counterintuitive energy-difference conserving dynamics in stark contrast to the standard Hermitian dynamics keeping the system's total energy constant. Therefore, we experimentally confirm unique properties of APT-symmetric systems, and further development in other areas of physics may provide new wave-manipulation techniques and innovative device-operation principles.

[1] Department of Physics, Hanyang University, 222 Wangsimni-Ro, Seoul 04763, Korea. [2] School of Electrical Engineering and Computer Science, University of Ottawa, 800 King Edward Avenue, Ottawa, ON K1N 6N5, Canada. [3] Electronics and Telecommunications Research Institute, Daejeon 34129, Korea. These authors contributed equally: Youngsun Choi, Choloong Hahn, Jae Woong Yoon. Correspondence and requests for materials should be addressed to J.W.Y. (email: jaeong.yoon@gmail.com) or to S.H.S. (email: shsong@hanyang.ac.kr)

In spite of their non-conserving nature and extra complexities, non-Hermitian symmetries and associated dynamics have triggered unprecedented interest in open physical systems because of their novel properties and potential applications[1,2]. For example, development of the parity-time (PT) symmetry in optics have suggested new ways of controlling light propagation and confinement involving spontaneous symmetry-breaking transition and non-Hermitian singularities[3–8]. As a plausible variant of the PT symmetry, anti-PT (APT) symmetry has been treated in positive−negative index multilayers[9], optically dressed atom lattices[10], and rapid coherent photo-atomic transport experiments[11]. An APT-symmetric Hamiltonian $H^{(APT)}$ can be conveniently defined in terms of a PT-symmetric Hamiltonian $H^{(PT)}$ such that $H^{(APT)} = \pm iH^{(PT)}$[11]. Consequently, PT-symmetric-like eigensystem structures involving exceptional points (EP), spontaneous symmetry-breaking transition, and self-intersecting energy-spectral topology appear and they result in PT-APT conjugate phenomena such as refractionless propagation, flat total transmission bands, and continuous lasing spectra[9–11].

Within the context of non-conserving binary oscillator problems, a PT-symmetric system has a characteristic Hamiltonian

$$H^{(PT)} = \begin{bmatrix} \varepsilon + i\gamma & \kappa \\ \kappa & \varepsilon - i\gamma \end{bmatrix}. \tag{1}$$

$H^{(PT)}$ describes two equally tuned oscillators at energy level $\varepsilon$ and with their attenuation (or amplifying, equivalently) rates differing by $2\gamma$. For non-dissipative inter-oscillator coupling where $\kappa$ is purely real-valued, the PT symmetry is an essential consequence as the system is invariant under the simultaneous parity inversion (P) and gain-loss exchange (T) operations. See Fig. 1a for schematic illustration of this property. The APT-symmetric counterpart is described by a Hamiltonian of the form

$$H^{(APT)} = \begin{bmatrix} -\varepsilon + i\gamma & i\kappa \\ i\kappa & \varepsilon + i\gamma \end{bmatrix} \tag{2}$$

which implies two equally amplifying oscillators at an amplification rate $\gamma$ and with their energy level differing by $2\varepsilon$, as illustrated in Fig. 1b. Importantly, no explicit physical symmetry is found for this generic APT-symmetric system under the PT operation. Moreover, the environmental energy-exchange scheme is completely different from the PT-symmetric counterpart. This argument suggests that essential dynamics in APT-symmetric systems might be remarkably different from the PT-symmetric counterpart even though the relation $H^{(APT)} = \pm iH^{(PT)}$ implies mathematically indistinguishable eigensystem structures for $H^{(PT)}$ and $H^{(APT)}$ in principle. Therefore, a comprehensive study on the stationary and dynamic properties of a generic APT-symmetric system is presently of importance in search of interesting and useful interaction configurations from conceptually diverse, open-system physics domains.

Here, we experimentally implement a model circuit of a generic APT-symmetric system and observe both stationary and dynamic properties associated with an EP singularity, spontaneous symmetry-breaking transition, and pseudo-Hermitian vector-space properties. Using a resistively coupled amplifying LRC resonators, we realize an APT-symmetric electrical system permitting precise parametric controls. Spectral and time-domain measurements reveal inverse PT-symmetry breaking transition of eigenvectors, associated bifurcation of complex eigenvalues at an APT-symmetric EP, and unique energy-difference conserving dynamics in stark contrast to the conventional systems described by Hermitian Hamiltonians. Therefore, we experimentally confirm essential consequences of the APT symmetry and associated anomalous non-Hermitian properties. Importantly, we show that the non-Hermitian quantum mechanics reveals the underlying physics of the observed properties in an intuitive manner although specific features of such properties can be understood in the standard circuit theory.

## Results

**Spectral properties of an APT-symmetric circuit.** The proposed electrical circuit consists of two amplifying LRC resonators connected in parallel through a coupling resistor as shown in Fig. 2a. The system simulates the APT−symmetric environmental-interaction scheme with negative resistor units ($-R_1$ and $-R_2$) providing a gain mechanism and with a coupling resistor ($R_C$) as a loss mechanism. For $R_C = R_1 = R_2 = R$ and $C_1 = C_2 = C$, essential dynamics of the system is described by a Schrödinger-type equation $d|v\rangle/dt = -iH^{(APT)}|v\rangle$, where $H^{(APT)}$ is given by Eq. (2). Here, the state vector $|v\rangle$ is defined such that $[V_1\ V_2]^T \equiv 0.5[\exp(-i\omega_0 t)|v\rangle + \exp(i\omega_0 t)|v\rangle^*]$ with $\omega_0 = 0.5[(L_2C)^{-1/2} + (L_1C)^{-1/2}]$ being average uncoupled-resonance angular frequency. The Hamiltonian matrix elements are determined by $\varepsilon = 0.5[(L_2C)^{-1/2} – (L_1C)^{-1/2}]$, $\gamma = 0$, $\kappa = (2RC)^{-1}$. See Supplementary Note 1 for mathematical treatment based on the Kirchhoff's circuit laws. Therefore, a generic APT-symmetric Hamiltonian is readily realized in this simple model circuit.

We assemble an APT-symmetric model circuit with parameters $R = 400\ \Omega$, $C = 425$ nF, $L_1 = 1.46$ mH, $L_2$ (variable) $= 1.46{\sim}0.78$ mH, and $R_A = R_B = 10\ k\Omega$. These circuit parameters yield constant $\kappa = 0.468$ kHz, variable $\varepsilon$ in a range from 0 to 1.18 kHz, and variable $\omega_0$ in a range from 6.39 to 7.57 kHz. Using this configuration, we investigate stationary APT-symmetric properties by exciting the circuit with a harmonically oscillating source signal injected at $V_2$ and monitoring $V_1(t)$ in the time ($t$) domain. A characteristic resonance spectrum is obtained by taking a Fourier-transformed intensity $W_n(f) = |F[V_n(t)]|$ in the frequency ($f$) domain. Measured $W_1(f)$ spectrum as a function of the energy-detuning parameter $\varepsilon$ is shown in Fig. 2b. Here, we clearly notice a branch-point splitting of the resonance peak at a threshold point of $\varepsilon = \kappa$. Remarkably similar spectral effects are found for nonlinear coupled-oscillator systems associated with the

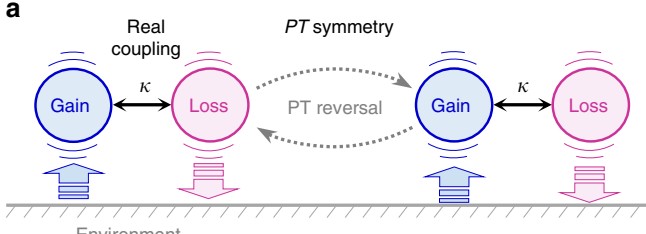

**a**

Real coupling    *PT* symmetry

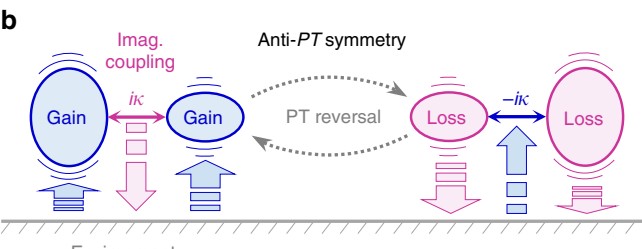

**b**

Imag. coupling    Anti-*PT* symmetry

**Fig. 1** Schematic diagrams of parity-time and anti-PT symmetric binary systems. **a** Parity-time (PT)-symmetric coupled oscillators implied by the PT-symmetric Hamiltonian $H^{(PT)}$ in Eq. (1). **b** Anti-PT (APT)-symmetric binary system derived from the relation $H^{(APT)} = \pm iH^{(PT)}$. In the two diagrams, $\kappa$ denotes inter-resonator coupling constant and vertical arrows indicate directions of energy exchange between the coupled-resonator system and environment

**a**

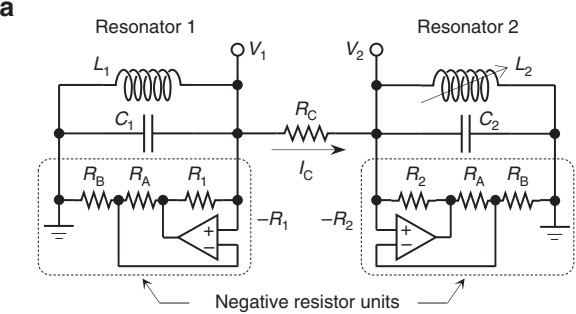

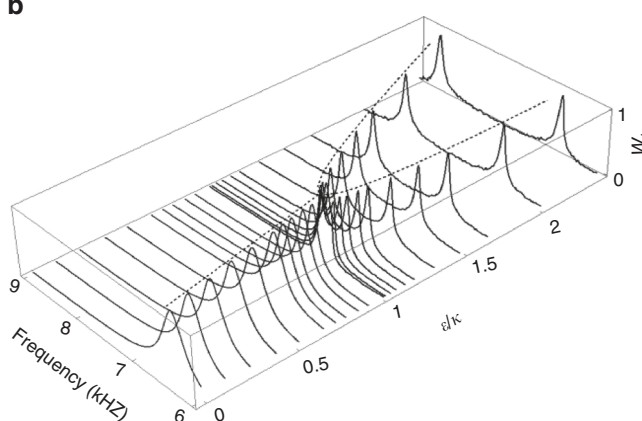

**Fig. 2** APT-symmetric model circuit and its spectral property. **a** Circuit diagram of APT-symmetric *LRC* resonators. The circuit consists of two resistively coupled amplifying *LRC* resonators with negative resistor units. A variable inductor is used for $L_2$ to precisely control the energy-detuning parameter $\varepsilon$. **b** $\varepsilon$-dependent resonance-excitation spectrum $W_1(f)$

injection-locking and pulling phenomena[12–15]. However, the resonance-peak coalesce observed in this case is obtained in a purely linear-gain regime and thereby it does not involve any nonlinear relaxation processes causing the conventional injection-locking phenomena. Instead, this property originates from an APT-symmetric EP and spontaneous symmetry-breaking transition as we will confirm in the following analyses in a quantitative manner.

Loci and bandwidths of the resonance peaks in the measured spectral profiles correspond to the real and imaginary parts of the eigenvalue

$$\lambda_{\pm} = \pm\sqrt{\varepsilon^2 - \kappa^2} \tag{3}$$

of $H^{(APT)}$. In the measured spectral profiles, the resonance-peak location and full-width at half-maximum (FWHM) in the frequency domain follow $f_{peak} = (2\pi)^{-1}[\omega_0 + \mathrm{Re}(\lambda_{\pm})]$ and $\Delta f_{peak} = \pi^{-1}[\delta + \mathrm{Im}(\lambda_{\pm})]$, respectively. Here, $\delta$ denotes a residual background absorption rate due to parasitic internal resistance in the constituent elements and its empirical value is around 25 Hz in our experiment. Therefore, $\mathrm{Re}(\lambda_{\pm})$ and $\mathrm{Im}(\lambda_{\pm})$ in experiment can be inferred from the peak location and bandwidth, respectively, as shown in Fig. 3a, b. Excellent agreement of the experimental values with the theory confirms that a generic APT-symmetric system is indeed realized in the model circuit with high degree of precision. In addition, a PT-APT conjugate property in the eigenvalue spectrum is evident therein. $\lambda_{\pm}$ shows a purely imaginary splitting for $\varepsilon < \kappa$, merging ($\lambda_{+} = \lambda_{-}$) at $\varepsilon = \kappa$, and a purely real splitting for $\varepsilon > \kappa$. Resulting from the fundamental relation of $H^{(APT)} = \pm iH^{(PT)}$, this characteristic is

exactly in parallel with the eigenvalue property near a PT-symmetric EP[2,16].

The threshold condition $\varepsilon = \kappa$ corresponds to the APT-symmetric EP where eigenvalues and eigenvectors simultaneously coalesce. In Fig. 3c, we show measured amplitude ratio $v_1/v_2$ of the state vector $|v\rangle$ at the resonance-center frequency as a function of $\varepsilon$. The amplitude ratio is uniquely defined for a state and the measured values clearly coalesce in the Gauss plane at $\varepsilon = \kappa$, confirming that this threshold condition represents an APT-symmetric EP. On top of the $v_1/v_2$ plot in Fig. 3c, we indicate ($v_1$, $v_2$) for the corresponding eigenvectors $|\lambda_{\pm}\rangle$. For $\varepsilon \leq \kappa$, both $|\lambda_{+}\rangle$ and $|\lambda_{-}\rangle$ are invariant under simultaneous $P$ (exchange of the two arrows each other) and $T$ (complex conjugation of the arrows) operations, i.e., the eigenvectors are in the exact PT-symmetry phase. In contrast, the PT symmetry in $|\lambda_{\pm}\rangle$ is broken for $\varepsilon > \kappa$. Therefore, the stationary response of the APT-symmetric circuit undergoes a spontaneous PT-symmetry breaking at the EP even though the system does not have any explicit physical symmetry as pointed out earlier.

In the eigensystem analysis so far, we have experimentally showed that a binary APT-symmetric EP involves a spontaneous PT-symmetry breaking in the eigenvector configuration and, in addition, the real/imaginary eigenvalue splitting property is reversed with respect to the PT-symmetric counterpart, i.e., real-eigenvalue splitting for the broken PT-symmetry phase ($\varepsilon > \kappa$) and imaginary-eigenvalue splitting for the exact PT-symmetry phase ($\varepsilon < \kappa$). In this respect, APT-symmetric binary systems and associated phenomena can be treated in a manner similar to the PT symmetry as far as their stationary responses are treated.

**Dynamic properties of an APT-symmetric circuit**. We further study dynamic properties of a generic APT-symmetric system in our resistively coupled LRC resonators. In particular, we investigate temporal responses for the broken PT-symmetry phase where the eigenvalue splitting is purely real-valued and the system's time-evolution does not involve a measurement-instability problem due to rapid exponential growth of the probe-voltage signals. The experimental procedures include following steps in the temporal order: Isolation of resonator 2 with the remaining parts including the coupling resistor $R_C$ and resonator 1; connecting the $V_1$ terminal to the ground to set $V_1 = 0$; time-harmonic excitation of resonator 2 at the resonance-center frequency by gain-assisted self-oscillation; disconnection of the $V_1$ terminal off the ground; connection of resonator 2 with the remaining parts; and acquiring $V_1(t)$ and $V_2(t)$ to determine the dynamic state $|v(t)\rangle = [v_1(t)\ v_2(t)]^T$ evolved from the initial state $|v(0)\rangle = [0\ 1]^T$. We conduct this experiment for $\varepsilon = 1.48\kappa$ and the result is summarized in Fig. 4a. Measured electric energy $E_n = 0.5 C_n V_n^2$ in capacitor $C_n$, magnetic energy $M_n = 0.5 L_n I_n^2$ in inductor $L_n$, and total energy $T_n = E_n + M_n$ for resonator $n$ are plotted as functions of time. An unprecedented property revealed in the measured time-domain response is beating patterns that conserve the energy difference $\Delta T = T_2 - T_1$. This is in stark contrast to the standard Hermitian dynamics keeping the system's net energy $T_1 + T_2$ constant and also to the PT-symmetric dynamics conserving a cross-conjugate product $v_1 v_2^* + v_1^* v_2$.

Within the context of the classical circuit theory, this unusual property is explained as originating from a specific configuration of the circuit and associated energy-variation rate properties. Configuring the circuit for the APT-symmetric environmental energy-exchange scheme in Fig. 1b, we required a set of circuit constant conditions such that $R_C = R_1 = R_2 = R$ and $C_1 = C_2 = C$. These conditions result in the resonator's energy-variation rate

$$\frac{dT_n}{dt} = \frac{V_1 V_2}{R} \tag{4}$$

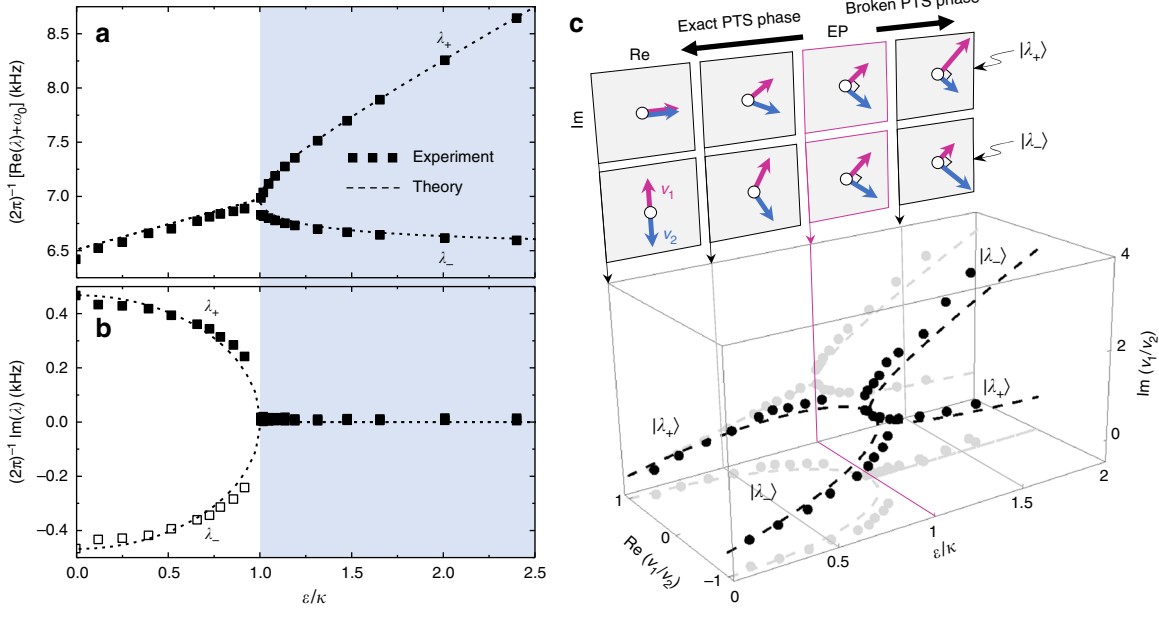

**Fig. 3** Eigensystem structure of an APT-symmetric circuit. **a, b** Real and imaginary eigenvalues inferred from the resonance-excitation spectra in Fig. 2b in comparison with theory. The theoretical curves are obtained by a binary Hamiltonian model derived from Kirchhoff's circuit laws. The shaded area for $\varepsilon/\kappa$ >1 indicates the parametric region of the broken PT-symmetry (PTS) phase. **c** State amplitude ratio $v_1/v_2$ as a function of the energy-detuning parameter $\varepsilon$, where $v_n$ denotes complex amplitude of the state vector $|v\rangle$ such that $|v\rangle = [v_1\ v_2]^T$. Circles indicate experimentally obtained values and dashed curves are obtained from the binary Hamiltonian model. The upper inset panels show corresponding eigenvector $|\lambda_\pm\rangle$ coordinates on the Gauss plane for $\varepsilon = 0$, $0.5\kappa$, $\kappa$, and $1.5\kappa$

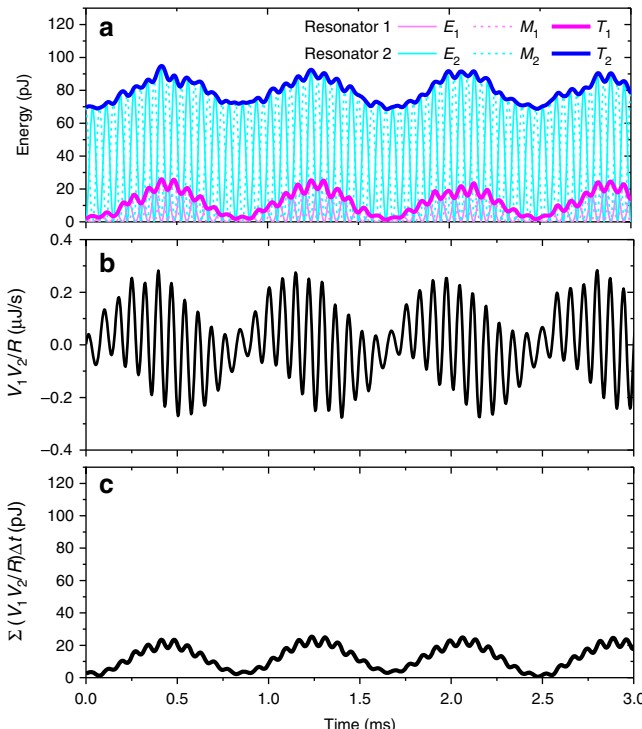

**Fig. 4** Energy-difference conserving dynamics in time domain. **a** Measured time evolution of the total energy $T_n$, electric energy $E_n$, and magnetic energy $M_n$ for an initial state $|v(0)\rangle = [0\ 1]^T$. We set $\varepsilon = 1.48\kappa$ in this measurement. **b** Measured voltage-product energy rate $V_1V_2/R$. **c** Integrated voltage-product energy rate $\Sigma(V_1V_2/R)\Delta t$ showing an exact correlation with the beating patterns in $T_n$. Here, the sampling time interval $\Delta t = 4\ \mu s$

which is essentially identical for the two resonators because the voltage-product energy rate $R^{-1}V_1V_2$ on the right-hand side of Eq. (4) is independent of the resonator index $n$. See Supplementary Note 2 for details. The energy-difference conserving dynamics implies that there is no net energy exchange between the two coupled resonators. In conventional coupled-resonator systems following the Hermitian dynamics, beating is a natural consequence of an inter-resonator energy exchange. In this respect, the specific beating patterns appearing in Fig. 4a for the $T_n$ profiles are not explained by the standard interpretation.

The counterintuitive beating pattern in the APT-symmetric circuit is induced by a periodic energy exchange between the whole resonator system and the environment. According to Eq. (4), the resonator's energy amplification or dissipation is determined by the sign of the voltage-product $V_1 \cdot V_2$. Therefore, the specific beating patterns in Fig. 4a are understood by the periodic change of the $V_1V_2$-product sign for the two resonators oscillating at slightly different frequencies. Note that the electric current $I_C$ through the coupling resistor $R_C$ is high for $V_1V_2 < 0$ and the Ohmic dissipation exceeds the system's net gain, resulting in attenuation of the resonant excitation. In contrast, $I_C$ for $V_1V_2$ > 0 is low and the response of the system is led by the gain that renders the excitation stronger. As key evidences, we provide measured voltage-product energy rate $R^{-1}V_1V_2$ in Fig. 4b and its integrated profile $J(t_a) = \Sigma_b [R^{-1}V_1(t_b)V_2(t_b)]\Delta t$ in Fig. 4c, where the dummy sampling-time index $b$ is running over 0 to $a$. The beating pattern in $J(t)$ in Fig. 4c shows a quantitative agreement with the $T_n(t)$ patterns in Fig. 4a, confirming the validity of the energy-variation rate relation given by Eq. (4).

In a more fundamental viewpoint, the energy-difference conserving dynamics is associated with a bi-orthogonal vector-space property. Following the pseudo-Hermitian representation of quantum mechanics[17] for our case with an APT-symmetric

Hamiltonian $H^{(APT)}$, the inner-product that permits a probabilistic description of state vectors should take a form $(\psi,\phi)=\langle\psi|\eta|\phi\rangle$, where $\eta$ is a Hermitian metric operator satisfying $\eta^{-1}H^{(APT)\dagger}\eta=H^{(APT)}$. In our case with a binary $H^{(APT)}$, the metric operator is given by

$$\eta=\begin{pmatrix}1 & 0\\ 0 & -1\end{pmatrix}\qquad(5)$$

and it yields a conserved quantity

$$\langle v|\eta|v\rangle = |v_1|^2 - |v_2|^2 = \frac{2}{C}\left[\langle T_1\rangle - \langle T\rangle_2\right],\qquad(6)$$

where $\langle\cdots\rangle$ indicates a time average of its argument over an oscillation cycle at $\omega_0$. Here, we use a relation $\langle T_n\rangle = 0.5\,C|v_n|^2$ implied from the definition of $v_n$ (see Supplementary Note 1). Therefore, the energy-difference conserving time-evolution is understood as a general property that applies to any APT-symmetric system regardless of system's details.

## Discussion

In conclusion, we have experimentally demonstrated an electrical circuit that simulates a generic APT-symmetric system. Stationary and dynamic properties were investigated using a resistively coupled amplifying-LRC-resonator circuit where precise parametric control and time-resolved measurement are highly feasible. We experimentally observed an APT-symmetric EP, inverse PT-symmetry breaking transition, and energy-difference conserving time-evolution as essential consequences of the APT symmetry. Although included in this paper is experimental confirmation of fundamental properties, our results propose the APT symmetry as a novel non-Hermitian interaction scheme where PT-symmetric-like eigensystem appears while associated dynamics is fundamentally different from standard Hermitian systems and even from the PT-symmetric counterpart. In particular, notion of the APT symmetry in optics and photonics is of great interest because APT-symmetric EPs and associated complex eigenvalue-splitting properties may provide new ways of creating EP-related phenomena such as unidirectional or non-reciprocal states of light[4–6], mode selection by a spontaneous symmetry-breaking transition[18], virtually diverging parametric sensitivity[19,20], and anti-adiabatic topological time-asymmetry[21,22].

In this consideration, it is worth mentioning that the mathematical distinction between PT and APT-symmetric systems is semantic in the framework of the pseudo-Hermitian quantum mechanics[17]. In addition, there is a wide variety of pseudo-Hermitian Hamiltonians that can be studied in substantially simplified experimental configurations by means of appropriate transformations. For example, a binary APT-symmetric Hamiltonian $H^{(APT)}$ is transformed into a PT-symmetric Hamiltonian $H^{(PT)}$ by a similarity transformation with a unitary operator

$$U=\frac{1}{\sqrt{2}}\begin{pmatrix}1 & 1\\ 1 & -1\end{pmatrix}\qquad(7)$$

implying that consequences of the PT symmetry might be equivalently studied in APT-symmetric systems or vice versa, depending on relative feasibility in experiments. Therefore, various PT-symmetric effects have a one-to-one correspondence to the APT-symmetric counterparts and this property could be used for generating novel non-Hermitian systems and devices with a comprehensive physical foundation. Further study of significant interest within this context is to realize time-varying APT-symmetric systems enabling topological operations around an EP[23,24]. This unique non-Hermitian interaction scheme was recently established for PT-symmetric EPs in the microwave-

transmission[21] and opto-mechanical oscillator[22] experiments. The subject is presently attracting a special attention because of its potential for robust time-asymmetric or nonreciprocal devices. In the potential APT-symmetric circuit configurations, required time-varying topological operations with an APT-symmetric EP can be readily created by introducing electrical tunability in the resistor and capacitor elements with appropriate transistors and varactor diodes. Furthermore, realization of optical systems involving mathematically identical non-Hermitian Hamiltonians should be feasible in coupled resonators or integrated-waveguide systems where required complex-index profiles can be effectively generated using impurity doping, functional thin films, and photonic nanostructures[3–8].

**Data availability**. The data that support the findings of this study are available from the corresponding authors on request.

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

## Acknowledgements

This research was supported in part by the Global Frontier Program through the National Research Foundation of Korea funded by the Ministry of Science, ICT & Future Planning (NRF-2014M3A6B3063708), the Basic Science Research Program (NRF-2018R1A2B3002539), and the Presidential Post-Doc Fellowship Program (NRF-2017R1A6A3A04011896).

## Author contributions

Y.C., C.H., J.W.Y., and S.H.S. conceived the original concept and initiated the work. Y.C. and J.W.Y. developed the theory and model. C.H. and Y.C. performed experiment. Y.C., C.H., and J.W.Y. analyzed the theoretical and experimental results. All authors discussed the results. J.W.Y. and Y.C. wrote the manuscript. Y.C., C.H., and J.W.Y. contributed equally to this work.

## Additional information

**Competing interests:** The authors declare no competing interests.

