## [Peer Review File · Nature Communications]

Reviewers' comments:

Reviewer #1 (Remarks to the Author):

The major claims of this paper are a comprehensive time- and frequency-domain study of an APT-symmetric system through an experiment using an electrical coupled-oscillator circuit.

This reviewer has the following comments for the authors:

- The coupled oscillator system that is being studied as a generic APT-symmetric system has been extensively studied in various communities, including circuits, mathematics etc. It is well known that when the natural frequencies of the oscillators are close to each other (i.e. within a locking range), the oscillators will lock to a common oscillation frequency. When the frequencies differ by more than the locking range, then the oscillators will not be locked and will exhibit an interesting time-domain evolution. The spontaneous symmetry-breaking transition described in Fig. 2 seems to be consistent with this well known understanding. Can the authors clarify if there any additional insights beyond this well-understood behavior?

- The energy difference conservation is interesting. However, it is difficult to appreciate the significance of this insight, since the reviewer cannot see any real application for the electrical coupled-oscillator experiment. Perhaps there are applications in other physical domains e.g. optics, but this paper fails to motivate these applications.

Reviewer #2 (Remarks to the Author):

Dear Editor

The paper introduces a bifurcation due to EP in eigensystem structure of anti-PT (APT) electric circuit. To the best of my knowledge this is the first time that such a phenomenon has been demonstrated in an experiment. Therefore, I recommend publication after addressing the following points:

1) Explain that the distinguishment between PT and APT is semantic since both cases are only examples for pseudo-hermitian Hamiltonians where $H = \eta^{-1} [H^{\text{deg}}] \eta$ and the conserved quantity is $\langle \eta \rangle$.

2) Emphasize that there are pseudo-hermitian Hamiltonians that are not PT but can be studied by a simple experimental setups (as electric circuit in their case). In such cases by simple transformations the observation of the effects of EP on the spectrum and on the dynamics of PT systems can be easily obtained from the study of APT systems, and vice versa.

3) Discuss the possibility to vary another experimental parameter beside epsilon/kappa such that by varying both of them in time to encircle the EP a time asymmetric switch would be obtained, as recently observed in microwave experiments, in Nature, 537(7618), 76-79 (2016), based on a theory first introduced in J. Phys. A: Math. Theor. 44 (2011) 435302 and theoretically demonstrated for molecular systems in Phys. Rev. A 88, 010102(R) (2013). Perhaps by lifting the requirement $R_1 = R_2$ and $C_1 = C_2$ they can produce such a switch.

Best

Nimrod Moiseyev

Author Response Letter to the Reviewer Comments

We hereby submit a revised manuscript as recommended in the previous decision letter. We thank the editor and reviewers for their dedicated time and consideration to review our manuscript. In response to the referee and editor comments, we have substantially improved our manuscript as delineated below. Therein, please note that the reviewer comments are in **Arial typeface in blue** while our responses are in Times New Roman typeface in black. In addition, all changes in the revised manuscript are highlighted in yellow.

Reviewer #1 (Remarks to the Author):

The major claims of this paper are a comprehensive time- and frequency-domain study of an APT-symmetric system through an experiment using an electrical coupled-oscillator circuit.

This reviewer has the following comments for the authors:

Comment 1) The coupled oscillator system that is being studied as a generic APT-symmetric system has been extensively studied in various communities, including circuits, mathematics etc. It is well known that when the natural frequencies of the oscillators are close to each other (i.e. within a locking range), the oscillators will lock to a common oscillation frequency. When the frequencies differ by more than the locking range, then the oscillators will not be locked and will exhibit an interesting time-domain evolution. The spontaneous symmetry-breaking transition described in Fig. 2 seems to be consistent with this well known understanding. Can the authors clarify if there any additional insights beyond this well-understood behavior?

Response 1) We thank Reviewer #1 for bringing our careful attention to this interesting point. Considering responses appearing in the frequency domain, remarkable similarity is indeed found between the resonance-center coalesce in our case and the injection-locking phenomena. Nevertheless, their underlying physics are fundamentally different. The injection locking phenomena are pertaining to nonlinear dynamics where different frequency components interact with each other. The injection locking of a system is accomplished when a stable stationary state (normal mode) dominates the system's excitation over other unstable states under the influence of amplitude-limiting or saturated-gain-type nonlinearities [R1–R4]. In this process, coupled oscillators unevenly oscillating in the beginning relax at a most stable normal mode (locking mode) over a certain period of time passage as described by an Adler-type nonlinear equation [R1,R2]. Meanwhile, other unstable normal modes in general oscillating at different frequencies are gradually attenuated.

In total contrast, the resonance-center coalesce in our case is purely a linear property. In our case, different normal modes in the broken PT-symmetry phase have a single

Table T1 | Comparison between the injection locking and the resonance-center coalesce in our case.

	Injection locking	Spontaneous symmetry-breaking transition
Spectral property requirement	Free-running frequency difference Δf_0 should be well within the passband width $f_0/(2Q)$ of the oscillator such that $f_0/(2Q) \gg \Delta f_0$. See associated cases in [R1–R4] for example.	There is no such frequency difference-bandwidth restriction in principle. For example, at the exceptional point in Figs. 2 and 3, $\Delta f_0 = 468$ Hz while $f_0/(2Q) = 17$ Hz $\ll \Delta f_0$. Therefore, our case obviously violate the frequency-difference requirement of the conventional injection locking.
Nonlinearity	Amplitude-limiting nonlinearity of an instantaneous type should be included. This requires a characteristic time constant for the amplitude-limiting mechanism to be much shorter than the shortest possible beat cycle.	No nonlinear interaction property is necessarily involved as a purely linear interaction property.
Normal-mode property in the single-freq. range	In a binary system, there are two solutions in the locking condition in general. These solutions do not necessarily have a single common frequency. The locking is accomplished when the stable solution dominates the system's response over the influence of an unstable mode [R4].	In the exact-PTS phase, the two normal-mode (eigenstate) solutions have a single common frequency as a natural consequence of their symmetry [R5]. Therefore, they do not need any additional mechanism such as the Adler-type phase-difference-relaxation process in the passage of time.
Threshold	Locking range is proportional to the ratio of injected signal to the oscillator excitation strength. There are two solutions: One is a stable locking mode and the other is unstable.	The exceptional point (EP) threshold is completely determined by signal-independent constants in the given oscillator configuration. In addition, there is a single normal-mode solution at the EP.

common frequency as a solution of a linear Schrödinger-type equation. Therefore, merging of two resonance peaks into a single peak is an immediate consequence, with no need of any nonlinear relaxation processes required for eliminating responses from parasitic unstable modes. Please note that the eigenvalue coalesce and associate eigenvector properties in our case quantitatively agree with a theory based on the linear Schrödinger-type equation as presented in Fig. 3. Consequently, our case is distinguished from conventional cases of the injection locking/pulling phenomena in major aspects as also summarized in Table T1.

Making clear this point in the revised manuscript, we have included additional statements in the second paragraph on page 2 as attached below.

“We assemble an APT-symmetric model circuit with parameters $R = 400 \Omega$, $C = 425 \text{ nF}$, $L_1 = 1.46 \text{ mH}$, L_2 (variable) $= 1.46 \text{ mH} \sim 0.78 \text{ mH}$, and $R_A = R_B = 10 \text{ k}\Omega$. These circuit parameters yield constant $\kappa = 0.468 \text{ kHz}$, variable ε in a range from 0 to 1.18 kHz, and variable ω_0 in a range from 6.39 kHz to 7.57 kHz. Using this configuration, we investigate stationary APT-symmetric properties by exciting the circuit with a harmonically oscillating source signal injected at V_2 and monitoring $V_1(t)$ in the time (t) domain. A characteristic resonance spectrum is obtained by taking a Fourier-transformed intensity $W_n(f) = |F[V_n(t)]|$ in the frequency (f) domain. Measured $W_1(f)$ spectrum as a function of the energy-detuning parameter ε is shown in Fig. 2b. Here, we clearly notice a branch-point splitting of the resonance peak at a threshold point of $\varepsilon = \kappa$. Remarkably similar spectral effects are found for nonlinear coupled-oscillator systems associated with the injection locking and pulling phenomena^{12–15}. However, the resonance-peak coalesce observed in this case is obtained in a purely linear-gain regime and thereby it does not involve any nonlinear relaxation processes causing the conventional injection-locking phenomena. Instead, this property originates from an APT-symmetric EP and spontaneous symmetry-breaking transition as we will confirm in the following analyses in a quantitative manner.”

Inclusion of these additional statements accompanies additional reference papers [12–15] and they corresponds to [R1–R4] in the same order.

References in Response 1

- [R1] R. Adler, *Proc. IRE* **34**, 351 (1946).
- [R2] L. J. Paciorek, *Proc. IEEE* **53**, 1723 (1965).
- [R3] B. Razavi, *IEEE J. Solid-State Circuits* **39**, 1415 (2004).
- [R4] A. Mirzaei and H. Darabi, *IEEE J. Solid-State Circuits* **49**, 360 (2014).
- [R5] A. Mostafazadeh, *Int. J. Geom. Methods Mod. Phys.* **07**, 1191 (2010).

Comment 2) The energy difference conservation is interesting. However, it is difficult to appreciate the significance of this insight, since the reviewer cannot see any real application for the electrical coupled-oscillator experiment. Perhaps there are applications in other physical domains e.g. optics, but this paper fails to motivate these applications.

Response 2) We agree that it is important to properly describe potential applications and specific future direction of our results in the revised manuscript. In this aspect, this comment is closely related to **Comment 5** by Reviewer #2. Please see associated text revision in **Response 5**. Therein, we explain application and further development of our results for realizing topological operations around APT-symmetric EP in similar electrical circuits and potential optical systems.

Reviewer #2 (Remarks to the Author):

Dear Editor

The paper introduces a bifurcation due to EP in eigensystem structure of anti-PT (APT) electric circuit. To the best of my knowledge this is the first time that such a phenomenon has been demonstrated in an experiment. Therefore, I recommend publication after addressing the following points:

Comment 3) Explain that the distinguishment between PT and APT is semantic since both cases are only examples for pseudo-hermitian Hamiltonians where $H = \eta^{-1}[H^{\text{deg}}]\eta$ and the conserved quantity is $\langle \eta \rangle$.

Response 3) We thank Reviewer #2 for this comment. This point is also closely related to **Comment 4** by Reviewer #2. In order to appropriately address **Comments 3** and **4**, we have inserted additional associated statements in the last paragraph of the main text on page 4 as also attached below.

“In this consideration, it is worth mentioning that the mathematical distinction between PT and APT symmetric systems is semantic in the framework of the pseudo-Hermitian quantum mechanics¹⁷. In addition, there is a wide variety of pseudo-Hermitian Hamiltonians that can be studied in substantially simplified experimental configurations by means of appropriate transformations. For example, a binary APT-symmetric Hamiltonian $\mathbf{H}^{(\text{APT})}$ is transformed into a PT-symmetric Hamiltonian $\mathbf{H}^{(\text{PT})}$ by a similarity transformation with a unitary operator

$$\mathbf{U} = \frac{1}{\sqrt{2}} \begin{pmatrix} 1 & 1 \\ 1 & -1 \end{pmatrix}, \quad (7)$$

implying that consequences of the PT symmetry might be equivalently studied in APT-symmetric systems or vice versa, depending on relative feasibility in experiments.”

Comment 4) Emphasize that there are pseudo-hermitian Hamiltonians that are not PT but can be studied by a simple experimental setups (as electric circuit in their case). In such cases by simple transformations the observation of the effects of EP on the spectrum and on the dynamics of PT systems can be easily obtained from the study of APT systems, and vice versa.

Response 4) Considering close connection of this comment to **Comment 3** by Reviewer #2, we have addressed this issue already in **Response 3**.

Comment 5) Discuss the possibility to vary another experimental parameter beside epsilon/kappa such that by varying both of them in time to encircle the EP a time asymmetric switch would be obtained, as recently observed in microwave experiments, in Nature, 537(7618), 76-79 (2016), based on a theory first introduced in J. Phys. A: Math. Theor. 44 (2011) 435302 and theoretically demonstrated for molecular systems in Phys. Rev. A 88, 010102(R) (2013). Perhaps by lifting the requirement $R_c=R_1=R_2$ and $C_1=C_2$ they can produce such a switch.

Response 5) We thank Reviewer #2 for leading us to this nice point regarding potential application or future direction of our results. In the revised manuscript, we have included additional statements at the end of the main text on page 4 as attached below.

“Therefore, various PT-symmetric effects have a one-to-one correspondence to the APT-symmetric counterparts and this property could be used for generating novel non-Hermitian systems and devices with a comprehensive physical foundation. Further study of significant interest within this context is to realize time-varying APT-symmetric systems enabling topological

operations around an EP^{23,24}. This unique non-Hermitian interaction scheme was recently established for PT-symmetric EPs in the microwave-transmission²¹ and opto-mechanical oscillator²² experiments. The subject is presently attracting a special attention because of its potential for robust time-asymmetric or nonreciprocal devices. In the potential APT-symmetric circuit configurations, required time-varying topological operations with an APT-symmetric EP can be readily created by introducing electrical tunability in the resistor and capacitor elements with appropriate transistors and varactor diodes. Furthermore, realization of optical systems involving mathematically identical non-Hermitian Hamiltonians should be feasible in coupled resonators or integrated-waveguide systems where required complex-index profiles can be effectively generated using impurity doping, functional thin films, and photonic nanostructures³⁻⁸.”

Therein, the additional suggested reference papers [J. Phys. A: Math. Theor. 44 (2011) 435302] and [Phys. Rev. A 88, 010102(R) (2013)] are included as [23] and [24], respectively.

Finally, we thank the referees and editor for encouraging us to substantially improve our manuscript with their substantive comments and kind suggestion.

REVIEWERS' COMMENTS:

Reviewer #1 (Remarks to the Author):

The modifications of the authors sufficiently address my concerns.

Reviewer #2 (Remarks to the Author):

The authors modified the paper according to my comments

Well done

Nice piece of work

Recommend to accept for publication

In response to the referee and editor comments, we have revised our manuscript as delineated below. Please note therein that the reviewer comments are in **Arial typeface in blue** while our responses are in Times New Roman typeface in black.

Reviewer #1 (Remarks to the Author):

The modifications of the authors sufficiently address my concerns.

Reviewer #2 (Remarks to the Author):

The authors modified the paper according to my comments. Well done

Nice piece of work

Recommend to accept for publication

Author response We sincerely thank Reviewers #1 and #2 for their time and consideration to review our revised manuscript again. In particular, we believe that their comments in the previous review reports indeed lead us to substantially improve our paper in major aspects.